# Focus Groups to Inform User-Centered Development of an eHealth Sleep Intervention for Adolescents: Perspectives of Youth with Insomnia Symptoms, with and without Pain

**DOI:** 10.3390/children10101692

**Published:** 2023-10-16

**Authors:** Michelle Tougas, Gabrielle Rigney, Christine Chambers, Isabel Smith, Joshua Mugford, Laura Keeler, Malgorzata Rajda, Penny Corkum

**Affiliations:** 1Department of Psychology and Neuroscience, Dalhousie University, Halifax, NS B3H 4R2, Canadag.rigney@cqu.edu.au (G.R.); christine.chambers@dal.ca (C.C.); isabel.smith@iwk.nshealth.ca (I.S.); mugfordjosh94@gmail.com (J.M.); laura.keeler@dal.ca (L.K.); 2Appleton Institute, School of Health, Medical and Applied Sciences, CQUniversity, Adelaide, SA 5034, Australia; 3Department of Pediatrics, Dalhousie University, Halifax, NS B3H 4R2, Canada; 4Department of Psychiatry, Dalhousie University, Halifax, NS B3H 4R2, Canada; malgorzata.rajda@nshealth.ca

**Keywords:** adolescent, sleep, pain, user-centred design, eHealth intervention

## Abstract

Introduction: Adolescence is a developmental stage that often coincides with increasing sleep problems. Focus groups were conducted to inform development of an adolescent eHealth sleep intervention by exploring opinions about (1) healthy sleep practices, and (2) using an eHealth intervention. Methods: Adolescents 14–18 years old experiencing symptoms of insomnia based on the Insomnia Sleep Index, with and without recurrent pain, and associated stakeholders (i.e., parents, school personnel, and health care providers) were recruited. Across six online focus groups, 24 adolescents with insomnia participated (14 pain-free, 10 with recurrent pain; 10 male, 14 female). Across seven online focus groups, 22 stakeholders participated, including 8 parents, 9 school professionals, and 5 health care providers (10 male, 8 female). Using a content analysis, subthemes were induced from transcripts. Results: Most healthy sleep practices were perceived as reasonable for adolescents to implement, except avoiding technology before bed and using bedrooms only for sleep. Three primary barriers to sleep practices were identified, including a variable schedule due to lifestyle factors, technology at night, and academics interfering with sleep, and only in the pain group, the barrier related to pain was identified. Content addressing adolescent-specific barriers was considered important to include in a sleep intervention. Desirable eHealth components included interactive features, videos, audio, and pictures to present information. A common barrier to using an eHealth sleep intervention was the program feeling too academic, with accessibility of the sleep information and strategies as a common facilitator. Conclusions: This research represents the first step in a user-centered approach to developing an adolescent eHealth sleep intervention. These results provide insights from a range of perspectives on guiding adolescents to follow healthy sleep practices. Next, these findings will be integrated in the development of an eHealth intervention for adolescents with and without recurrent pain.

## 1. Introduction

One in four adolescents aged 11–18 report sleeping less than the recommended 8–10 h a night [1]. Adolescence is a developmental period in which changes to sleep often occur due to a range of biopsychosocial factors [2]. Adolescents experience a phase shift in circadian processes resulting in a physiological change that drives staying up later at night and sleeping later in the morning [3]. This is also a time of reduced parental involvement in regulating sleep schedules, as well as increased autonomy for activities that may contribute to adolescent sleep difficulties [4]. Adolescents continue to wake early for school while variable schedules and biological shifts in sleep times interfere with healthy sleep [5].

During adolescence, recurrent pain (at least once per week for at least 3 months) often begins and influences up to 30% of adolescents [6]. Whereas 25% of pain-free adolescents report sleep problems, these are reported in up to 50% of adolescents with recurrent pain [7,8]. The onset of recurrent pain can have broad negative impacts on academic, psychological, and physical functioning [9]. A bidirectional relationship between sleep and pain is suggested, wherein pain may interfere with sleep and sleep disturbances may aggravate pain [10,11]. While appropriate sleep duration and quality is challenging to achieve in adolescence, youth with recurrent pain are even more affected.

Inadequate sleep quality and quantity in adolescents have been linked to negative effects on psychological, academic, and physical functioning [12,13]. Moreover, daytime tiredness and night waking have been linked with the onset and aggravation of recurrent pain [14,15]. Considering the effects of poor sleep on functioning, increasing awareness of its importance and teaching healthy sleep practices are critical during this vulnerable time.

Healthy sleep practices include following recommendations about sleep behaviours and lifestyle factors (e.g., eating healthy, exercising) [16]. Adolescents have reported being aware of most healthy sleep practices, however, with difficulty successfully implementing them [17,18]. Several factors that adolescents report as preventing successful healthy sleep practices include available time, varying schedules, technology use, active minds, and a distracting bedroom [17,18,19].

Most adolescent sleep interventions have used in-person delivery providing healthy sleep practices and individual- or group-based cognitive behavioral therapy for insomnia, with consistent improvements in sleep hygiene, quality, and duration [20,21,22,23]. Only two studies have reported testing sleep interventions with recurrent pain adolescent populations, with reportedly improved insomnia symptoms, sleep habits, pain frequency, and duration [24,25]. In-person interventions are generally time-intensive, ranging between six and ten, 90–100 min, weekly sessions, with protocol compliance as a common barrier to success of adolescent behavioural sleep interventions [26]. While in-person interventions are generally effective for improving sleep outcomes, they are resource-intensive and difficult for adolescents to access and engage with [27].

Sleep education delivered in schools allows for access to large numbers of adolescents with content delivered during school hours in classroom-style formats [28]. However, while these interventions have shown improvement in sleep knowledge, impact on changing sleep behaviours has been inconsistent [29]. Sleep education programs typically focus on education rather than behavioural strategies, and do not include options for tailored treatment or specialized information such as the link between pain and sleep.

The implementation of eHealth interventions can help to address engagement and accessibility, with online interventions allowing for tailored and personalized content. User input is a key component in developing desirable and engaging eHealth interventions [30]. While in-person interventions can be tailored in real-time, eHealth applications need to consider user engagement in advance. Despite the potential for eHealth to overcome barriers that interfere with access to effective health interventions, the majority of available health care applications do not follow a user-centered design, and fail to involve users in the development process [31]. End-users involved in early intervention design and testing can provide consideration of factors that encourage or deter use. Understanding user needs throughout conceptualization increases the likelihood of developing a product that will meet user needs and guide improvement in health outcomes [30].

With adolescents’ widespread use of the internet, interventions provided online offer cost-effective and timely delivery methods [32,33]. Despite these advantages, only three online adolescent sleep interventions have been published, all reporting positive changes in sleep outcomes across a range of sleep variables (e.g., sleep efficiency) and measures [22,34,35]. Two studies reported including adolescent end-users in the development process; however, one had a sample that included only 24% of adolescent participants 15–18 years old (the remaining sample being 19–24 years old, [34]), and the other reported that only one-third of participants completed the entire intervention during pilot testing [35]. More research is needed for further intervention development and testing to meet adolescent needs for eHealth sleep treatment delivery.

This current paper describes the first step of a user-centered process of developing an eHealth sleep intervention for adolescents with sleep problems, with and without recurrent pain. The primary aim of this research was to gather input through focus groups from adolescents and stakeholders (i.e., parents, educators, health care professionals) on (1) healthy sleep practices that are perceived as either reasonable or difficult to implement, as well as barriers and facilitators to following healthy sleep practices, and (2) desired content, features, and visual presentation of an eHealth sleep intervention, as well as barriers and facilitators to using an eHealth intervention. The secondary aim of this research was to compare responses between adolescents and stakeholders, and between pain and pain-free participants, to identify potential group-based differences.

## 2. Materials and Methods

### 2.1. Participants

Adolescent participants were eligible if they were between the ages of 14 and 18 years, experienced at least mild insomnia problems on the Insomnia Severity Index (see below), lived in Canada for at least 6 months, were enrolled in a junior high or high school, were proficient in English, and had access to a computer with internet. Adolescents with pain were required to experience recurrent pain that occurred at least once per week for at least 3 months, with sleep problems perceived as related to pain.

Stakeholders were recruited into pain-free and pain groups based on their interaction with adolescents experiencing recurrent pain. Parents were eligible if they had a child who met the adolescent eligibility criteria (their child did not need to participate in this study). School professionals (i.e., teachers or school counsellors) were eligible if they were working with adolescents aged 14–18 years. Health care professionals were eligible if they were working in a Canadian clinical practice and had experience treating adolescents with recurrent pain. Adolescent (both pain and non-pain group participants) and stakeholder participants were excluded if they reported experiencing an intellectual disability, and/or visual or hearing impairment that would interfere with participation.

### 2.2. Procedure

Participants were recruited between March 2017 and September 2019 from across Canada through online advertisements, social media, mailing lists, posters, and word of mouth. Interested individuals completed an online eligibility questionnaire and consent form. All consenting participants were emailed a link for an online survey to gather demographic information (see below). We did not ask nor control for whether the participants knew each other (none reported this as a concern).

A synchronous online focus group design was followed for four groups: (1) adolescents without pain, (2) adolescents with pain, (3) stakeholders corresponding to adolescents without pain, and (4) stakeholders corresponding to adolescents with pain. As such, adolescents and stakeholders were not in the same focus groups. Focus groups were conducted using a secure online web conferencing software (Blackboard Collaborate Ultra, v17.1). Each 1.5 h focus group (all of which were in the afternoon or early evening) was led by two moderators (M.T., G.B., J.M., or L.K.) using a discussion guide that included semi-structured, open-ended questions. The questions focused on which healthy sleep practices adolescents perceived as reasonable or difficult to implement, barriers to following healthy sleep practices, and suggestions for content, features, and visual presentation of an eHealth intervention, as well as barriers and facilitators to using an eHealth intervention. Participants in the pain-related groups were additionally prompted about the role that pain may have in each topic. These pain-related prompts were not included for the pain-free groups. Focus groups were audio-recorded (with the conferencing software Blackboard Collaborate Ultra v17.1) and transcribed (by J.M., L.K., or M.T.). All participants were emailed an honorarium CAD 20.00 gift card.

### 2.3. Measures

#### 2.3.1. Questionnaires

##### Screening Questionnaires

Screening questionnaires assessed eligibility regarding adolescent age, insomnia symptoms, pain frequency, and school attendance. Stakeholders defined their role (e.g., parent, school professional, health care provider, and pain or pain-free discussion). All participants were screened for Canadian residency, English proficiency, access to a computer with internet, and cognitive or visual/hearing impairment.

Screening for sleep problems was conducted using the 7-item Insomnia Severity Index with scores ranging from 0 to 28, and a score of 8 or above endorsing at least mild insomnia symptoms [36]. The ISI shows high internal consistency (α = 0.90) [37].

##### Demographic Questionnaire

The demographic questionnaire asked participants to report age, sex, ethnicity, and geographical location in Canada. Adolescents were also asked to report pain type(s), average intensity (10-point numerical rating scale), and how long they had been experiencing pain. Stakeholders were not asked to report on insomnia or pain.

##### Sleep Hygiene Index

The Sleep Hygiene Index (SHI) is a 13-item scale that assesses sleep hygiene, also referred to as healthy sleep practices [38,39]. Scores range from 0 to 55, with higher scores representing poorer sleep hygiene. Based on a psychometric analysis, scores of 41 and above are in at least the 75th percentile [39] and for the purposes of this research were considered to be “poor”. The SHI has demonstrated acceptable internal consistency (α = 0.66) and reliability (*r* = 0.71, *p* < 0.01) [39]. This was used to describe the sample in terms of sleep habits, which is a separate construct from insomnia symptoms (as measured with the ISI).

### 2.4. Data Analysis

Quantitative data (e.g., demographic, sleep, and pain questions) were analyzed using IBM Corp. SPSS Version 24 software using descriptive statistics (i.e., frequency counts, percentages, means, standard deviations, and ranges). Unpaired t-tests were used to compare the means between pain and pain-free adolescent and stakeholder groups.

An inductive content analysis was followed [40] with two reviewers (M.T., L.K.) independently reviewing the transcribed data. Each reviewer identified subthemes within the main focus group questions when a topic was introduced by more than one participant. The reviewers discussed their themes to arrive at a consensus of what label and type of information should be applied to each subtheme. Reviewers discussed discrepancies until consensus was reached and revised the coding as needed. The reviewers continued until no new subthemes were identified and coding agreement of at least 80% was reached. Both reviewers then reviewed and coded each of the focus groups’ transcripts using the final coding guide. Discrepancies from the final coding were discussed with resolution by a third reviewer as needed (G.R., or P.C.). Coded transcripts were imported into a qualitative analysis program, NVivo 12.0, to organize and review the coded data.

## 3. Results

### 3.1. Participants

A total of 24 adolescents across six focus groups and 22 stakeholders across seven focus groups participated (see Figure 1 for participation flow). Focus groups ranged from two to nine participants. There were no significant differences between the pain-free and pain groups in terms of sex, age, ethnicity, ISI, or SHI (see Table 1 for participant demographics, and Table 2 for adolescent sleep and pain characteristics). Of the 14 adolescents who reported being pain-free at eligibility assessment, 6 reported experiencing recurrent pain when reporting demographics. Considering that pain levels were reportedly lower in the non-pain group, the self-selected pain-free group was maintained for these participants (Table 2).

### 3.2. Focus Group Themes

Focus groups discussed two topics based on the research objectives: (1) perception and use of healthy sleep practices, and (2) eHealth intervention development. These objectives were structured into specific questions asked across focus groups. See the focus group discussion guide in Appendix A.

#### 3.2.1. Perception and Use of Healthy Sleep Practices

Focus group moderators introduced several healthy sleep practices, see Table 3, and asked participants about (1.1) following healthy sleep practices, (a) sleep practices reasonable to implement, and (b) sleep practices most difficult to implement, and (1.2) barriers and facilitators, (a) barriers to following healthy sleep practices, and (b) facilitators to following healthy sleep practices.

#### 3.2.2. Following Healthy Sleep Practices

##### Sleep Practices Considered Reasonable to Implement

All types of participant groups were familiar with each of the recommended healthy sleep practices. The majority of practices were perceived as reasonable for adolescents to implement. The healthy sleep practice of engaging in calming activities before bed was discussed most often, with participants describing using many different calming strategies including reading, mindfulness, relaxation, drinking caffeine-free tea, listening to quiet music, snuggling with a pet, and coloring. Participants acknowledged that “*winding down is key*” and an adolescent identified “*for that half hour before bed I’m really relaxed and focusing on my reading. That’s helpful to get me ready to go to bed*”.

Next, following a bedtime routine was discussed most often, where many participants reported having a routine or being interested in starting one. Participants were aware of the helpfulness of routines, with one adolescent stating “*I know that routines really, really, do help because it kind of tells my body that it is time to go to sleep and makes me look forward to going to bed at night*”. Participants were also consistently aware of the connection between regular physical activity and healthy sleep, with one stakeholder identifying “*if they get active and exhausted, when they come home, they will crash and get that much needed sleep*”.

The other healthy sleep practices perceived as reasonable for adolescents to implement included eating a healthy diet, keeping the bedroom quiet and dark for sleep, and sleeping 8–10 h at night. Only pain-related adolescents and stakeholder groups discussed the importance of having no caffeine in the few hours before bed. Further, only adolescents with pain discussed knowing to avoid large meals before bed. Notably, however, these last two healthy sleep practices were discussed infrequently and were not identified by any participants as factors that would be difficult to change.

##### Sleep Practices Considered Difficult to Try or Change

Difficulty keeping technology out of the bedroom at night resulting in a negative influence on adolescents’ sleep was reported. One adolescent noted that “*I always use my phone before bed. I know most people my age do, I know it’s bad, and I don’t know if it is possible for me to not have in my room*”. The other healthy sleep practice that was identified as difficult to change was using the bedroom only for sleep. Stakeholders and adolescents acknowledged that the bedroom is where adolescents spend a lot of personal time, and it is difficult to expect them to use their room only for sleeping. One parent highlighted that “*it is important for teenagers to have a space that is their own where they can express themselves, for a lot of teenagers, their bedroom kind of becomes that sort of space*”. No group differences were identified for practices perceived as difficult.

#### 3.2.3. Following Healthy Sleep Practices

##### Barriers to Following Healthy Sleep Practices

Although participants identified that many healthy sleep practices felt manageable to try, they identified several factors that get in the way of being able to follow recommendations successfully and consistently. The most common barrier was inconsistent schedules due to lifestyle. Stakeholders and adolescents themselves highlighted that variable schedules due to school, extracurricular activities, social engagements, or employment interfere with consistent bedtimes or obtaining a full night’s sleep. One adolescent acknowledged that “*My extracurricular activities end at different times, so I have to go to bed later some nights. It is hard for me to find an exact time to go to bed*”. A parent reported that “*Sometimes it feels like they’re fighting a losing battle. It’s complicated to find a consistent bed and wake time that accommodates both a school and activity schedule*”.

The use of technology at night was the next most common barrier. Participants discussed using social media, watching videos, and playing video games before falling asleep. One teen highlighted “*I love going to bed late because I always chat with my friends and watch some YouTube videos. It’s not good for my health, but it’s like an addiction*”. Another common barrier to falling asleep was having an active mind. A health professional noted that “*bedtime is when they are together with their thoughts and if they haven’t had the opportunity to learn strategies to manage those thoughts, it can be overwhelming*”.

The barrier of motivation was discussed by mostly stakeholders. One parent indicated that “*it’s a really tricky age to try and enforce boundaries around things in your household, all you can do is give information and try to motivate them as best as you can*”. Academics interfering with sleep was discussed by mostly adolescents, with one adolescent highlighting that “*because school is so busy and important, you kind of prioritize school over sleeping*”. Only in the pain focus groups was pain mentioned as a barrier to following healthy sleep practices, with a health professional noting that “*pain-related fatigue can contribute to naps during the day or when they get home from school*”, and an adolescent reporting that “*pain is pretty much the only thing that gets in the way of my sleep*”.

##### Facilitators to Healthy Sleep Practices

Three facilitators to following healthy sleep practices were discussed. Participants described how knowledge about both the negative impacts of sleep problems and the benefits of healthy sleep can facilitate following healthier sleep practices. A health professional acknowledged that “*we need to tell them the benefits of going to bed and having decent sleep, and the impacts it will have on their future*”. Focus group discussions also highlighted the importance of modeling healthy sleep practices by parents, friends, or celebrities to motivate adolescents. One parent acknowledged that “*another good motivator is leading from example. If parents are following a healthy sleep schedule, then maybe the teenager will follow*”. Only stakeholders (and no adolescents) identified setting goals as a facilitator to healthy sleep practices, with a parent noting “*you somehow have to find a way to get them to realize and want to take back the control that they have over their lives*”.

### 3.3. eHealth Intervention Development

Focus group moderators discussed planning the development of an adolescent eHealth sleep intervention. Discussions explored eHealth intervention preferences, including (a) desired content, (b) appealing features and visual presentation, and (c) barriers and facilitators to use.

#### 3.3.1. eHealth Sleep Intervention Content

When participants discussed what content they perceived as most important to include in an eHealth sleep intervention, they provided suggestions from their own experiences and knowledge for how others can overcome barriers to specific healthy sleep practices. The most common suggestions were strategies to engage in regular physical activity, which included trying activities that are not sports, setting alarms to schedule activity, being active early in the day, and trying short bursts of exercise. Other strategies that participants identified as valuable to include in the intervention were suggestions for avoiding technology in the bedroom at night, including charging phones away from the bed or bedroom, shutting off the internet, and informing peers when turning off the phone. Finally, all types of focus group participants highlighted that the intervention should include recommendations for achieving consistent bedtimes and waketimes; however, the only specific suggestion was to set a daily alarm.

Only pain-related adolescent and stakeholder groups discussed ideas for tailoring content to recurrent pain, with most suggestions provided by health professionals. Suggestions included recommending moderate rather than strenuous physical activity, encouraging mild activity even when pain is present, pacing activities throughout the day to avoid an evening pain flare-up, being aware of pain medications that may interfere with sleep, and incorporating pain management strategies into a daily bedtime routine.

#### 3.3.2. eHealth Features and Visual Presentation

Participants highlighted the importance of including interactive features within an eHealth intervention. Interactivity suggestions included games, quizzes, fillable forms, and customizable personal bedtime or physical activity routines. Participants consistently identified that a reward system such as collecting points or meeting personal achievements would motivate engagement. A sleep diary was often recommended as a feature that would help participants track progress. The majority of these discussions were provided by adolescents with the exception of delivering an eHealth intervention as an app rather than a website, recommended by both stakeholders and adolescents. Almost all comments regarding visual presentation were also provided by adolescents, with a focus on minimizing text through the use of videos, audio, and pictures. Adolescents also recommended bright colors and a simple design for easy navigation.

#### 3.3.3. Barriers and Facilitators to Using eHealth Interventions

Participants consistently reported that having an intervention that feels academic or is too much work would deter adolescent use. Participants also discussed the importance of first impressions and how a negative first experience could turn adolescents away from the intervention. Only adolescents, and not stakeholders, reported that generic or stagnant information would stop them from returning to the intervention. Specifically, one adolescent stated “*If we end up getting similar material every day, I don’t think a lot of people would be drawn to it*”.

The only facilitator to using an eHealth intervention that was discussed across focus groups was accessibility. Participants agreed that having relevant information about sleep problems and healthy sleep practices all in one place would encourage using an eHealth intervention. One teen acknowledged that “*it saves people from looking about the internet for their information and if we have this program that it will all be there in one place*”.

## 4. Discussion

This paper describes the first step of a user-centered process for developing an eHealth sleep intervention for adolescents with sleep problems, with and without recurrent pain. Adolescents and stakeholders provided opinions through online focus groups that discussed healthy sleep practices and preferences for an eHealth sleep intervention. Most healthy sleep practices were considered to be reasonable for adolescents to implement, with the exception of avoiding technology before bed and using the bedroom only for sleep. However, many barriers to following sleep practices were identified, the most common being having a variable schedule. Participants recommended content to include within an eHealth sleep intervention, primarily being strategies suggested for overcoming barriers to following healthy practices. Components suggested to include in an eHealth intervention were interactive features, videos, audio, and pictures to present information. Participants identified a program appearing to be too academic as a common barrier to using an eHealth sleep intervention, with accessibility as a common facilitator. Most discussions were consistent across groups, with a small number of group-specific topics.

Participants were generally aware of most healthy sleep practices presented during the focus group [16], the majority being perceived as reasonable for adolescents to implement. Despite perceiving most practices as reasonable to implement, multiple barriers to following these in terms of implementation were identified. These results are consistent with those of other focus groups exploring adolescents’ perceptions of sleep behaviour, with common barriers to following healthy sleep practices across studies including time demands, variable schedules, use of technology, difficulty switching off their minds at bedtime, and a distracting bedroom environment [17,18]. These common barriers are therefore important to consider when developing an adolescent eHealth sleep intervention, with recommendations to overcome adolescent barriers provided in conjunction with teaching healthy sleep practices.

Participants in the current study were not aware of many facilitators of sleep behaviour change, aside from having information about the positive and negative impacts of sleep. This finding is distinct from existing research in which adolescents have identified parental involvement as a facilitator to following healthy sleep practices [19]. Provision of a sleep intervention during adolescence may help to buffer the shift away from parental support of sleep behaviours at this important developmental stage [4,41]. To facilitate self-directed change in sleep behaviours, it is therefore imperative that an eHealth sleep intervention integrates clear information about the negative impacts of poor sleep and the positive impacts of healthy sleep, particularly for healthy sleep practices that are considered to be difficult to change.

Participants provided suggestions for content, features, and presentation as well as barriers and facilitators to using an eHealth sleep intervention. The content perceived as most important to include was primarily suggestions about overcoming barriers to healthy practices (e.g., charging phones away from the bed to minimize technology use at night). This will need to be taken into consideration when developing the eHealth intervention, as the intervention program would be available on digital devices (e.g., smartphone app, computer) and if used close to bedtime, would go against recommendations for healthy sleep habits. Regarding features, the need for an interactive and engaging intervention was identified, with suggestions for activities like games or quizzes. Participants also recommended integrating reward systems and diary tracking to maintain engagement. For visual presentation, adolescents identified the need for minimal text, simple design, and information presented in bright colors over a range of formats including audio, video, and pictures. These results are consistent with previously reported adolescent focus groups addressing other health conditions that have also identified the importance of options for how an eHealth intervention is delivered [42]. Listening to feedback and integrating these suggestions into the content, features, and visual design of an eHealth sleep intervention are critical to create a program that will be used [43].

Participants identified that an accessible program with relevant information about sleep problems and healthy sleep practices contained in one place will facilitate using an eHealth intervention. A program feeling too educational or requiring a lot of work was a reported barrier to using an eHealth sleep intervention. Adolescents in previous research have also suggested using minimal text in eHealth intervention delivery [42,44]. Notably, in pilot testing of an app-based adolescent eHealth sleep intervention, only 33% of adolescents completed all components of the program, with the main reason for discontinuing cited as too much text and repetitive information [35]. Following end-user recommendations to create an accessible intervention will enhance the likelihood of adolescents overcoming barriers to using and adhering to an eHealth sleep intervention [45].

The secondary aim of this study was to compare responses and identify potential differences between adolescent and stakeholder participants, and between groups related to pain and pain-free discussion topics. Discussions about healthy sleep practices were mostly consistent across focus groups. When discussing barriers to following healthy sleep practices, adolescent participants identified schoolwork as a barrier, a concern that was not raised by many stakeholders. Stakeholders instead identified motivation as something that may be holding adolescents back, a topic that did not come up in the adolescent groups. Perhaps to address this barrier, only stakeholders further identified setting goals to facilitate motivation in adolescents to improve their sleep behaviour. In current sleep education literature, programs that have incorporated a motivational theoretical foundation have been found to be most successful in improving sleep behaviours [46]. Including motivational components (e.g., motivational interviewing techniques, feedback on success with the program) within an eHealth sleep intervention is important to implement despite this suggestion being identified by only stakeholders.

### Strengths and Limitations

While this study collected a range of perspectives from adolescents with and without pain and associated stakeholders, these results reflect participants’ opinions and not necessarily those of all adolescents and stakeholders. The adolescent samples were difficult to retain, with many individuals failing to attend focus groups despite confirming availability. This resulted in some focus groups that were smaller than the recommended minimum of three participants [47], which may have resulted in fewer suggestions than a larger group. Focus groups were held entirely online without any face-to-face contact, potentially influencing the sample composition by limiting participation to only those comfortable participating online. This online platform, however, allowed for recruitment of individuals Canada-wide, which would not have been possible if the focus groups were held in-person. The focus group guide was used to provide structure and in doing so may have limited the topics discussed as moderators did not ask about any additional topics. Although the focus groups did not follow a systematic approach for obtaining agreement on all subthemes from all participants, there was consistency across participant groups on most topics discussed. We also were limited in describing the sample as we did not ask for information on the participant’s location within Canada and as such do not know to which geographical regions we can generalize our findings. We also did not recruit an ethnically diverse sample, and so caution needs to be taken when generalizing the study’s findings. Further caution is recommended given that a sizable number of participants did not attend the groups despite confirming their interest and availability. This could lead to a sampling bias as only those participants who had the resources (such as personal responsibility or other factors) to attend the focus groups had a voice in this study, and their opinions may differ from those who did not attend the focus groups. Moreover, this lack of engagement with online mediums may suggest that there may be uptake and adherence issues with online interventions for adolescents. We also did not ask all possible questions about eHealth interventions (e.g., we did not ask about alternatives to this such as analog versions), as we wanted to ensure that the focus groups were not too long. Lastly, we only used prompts related to pain and sleep in the pain group. It was not expected that the non-pain group would report recurrent pain on the demographic form after having not reported this on the eligibility screening questionnaire. This means that we lost an opportunity to learn from those adolescents who had recurrent pain (albeit less severe) who were in the non-pain group.

## 5. Conclusions

This study was the first step in a user-centered approach to the development of an eHealth sleep intervention, through gathering opinions from adolescents with insomnia symptoms, some of whom had recurrent pain and others that did not, as well as stakeholders (parents, educators, and health care professionals). Results from these focus groups have provided a range of meaningful insights on how to guide adolescents to follow healthy sleep practices and how best to deliver that information through an eHealth intervention. The information gathered from these groups will be incorporated into an eHealth sleep intervention, with the aim of tailoring the program to enhance adolescent engagement. While these results are specific to the development of an eHealth sleep intervention, many of the recommendations could provide a foundation for eHealth interventions for other adolescent behavioural health concerns (e.g., eating a balanced diet, engaging in appropriate physical activity).

## Figures and Tables

**Figure 1 children-10-01692-f001:**
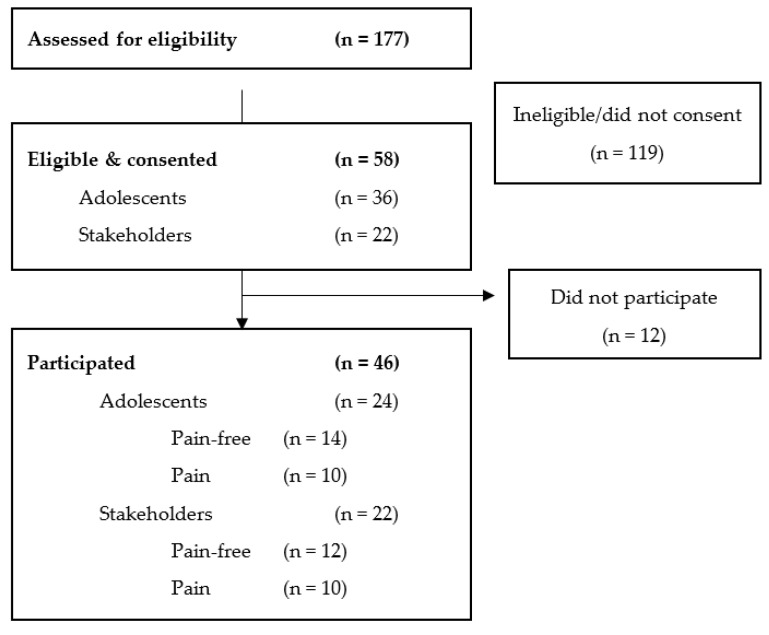
Focus group testing participant flow.

**Table 1 children-10-01692-t001:** Adolescent and stakeholder participant demographics.

Demographic Characteristics	Adolescents (*n* = 24)	Stakeholders (*n* = 22)
Age (years), mean (SD)	16.50 (1.25)	41.38 (10.70)
Sex		
Female	*N* = 14 (58%)	*N* = 12 (55%)
Male	*N* = 10 (42%)	*N* = 10 (45%)
Ethnicity		
Caucasian	*N* = 13 (54%)	*N* = 13 (59%)
South Asian	*N* = 4 (16%)	*N* = 5 (23%)
Other	*N* = 7 (30%)	*N* = 4 (18%)
Stakeholder Type		
Parents	--	*N* = 8 (36%)
School professionals	--	*N* = 9 (41%)
Health care professionals	--	*N* = 5 (23%)

**Table 2 children-10-01692-t002:** Adolescent sleep and pain characteristics.

	Adolescents without Pain, Mean (SD)	Adolescents with Pain, Mean (SD)
Insomnia Severity Index ^1^	19.8 (2.78)	17.29 (6.60)
Sleep Hygiene Index ^2^	39.9 (5.92)	37.10 (6.64)
Recurrent pain	*N* = 6	*N* = 14
Pain intensity (0–10)	4.50 (2.34)	5.38 (1.06)
Pain duration (months)	3.50 (4.04)	6.63 (4.62)
Pain Type		
Headache	*N* = 1	*N* = 10
Musculoskeletal pain	*N* = 2	*N* = 4
Abdominal pain	*N* = 1	--
Other	*N* = 2	--

^1^ Scores ranged from 0 to 28, and a score of 8 or above endorsed at least mild insomnia symptoms. ^2^ Scores of 41 or higher are considered ‘poor’.

**Table 3 children-10-01692-t003:** Healthy sleep practices introduced to all focus groups.

Label	Definition
Sleep duration	8–10 h of sleep each night, without any naps during the day
Consistent times	Going to bed and waking at the same time (within 1 h) every day, including weekends
Bedtime routine	Doing the same things in the same order around the same time, every night before bed
Caffeine	No caffeine in the few hours before bed
Exciting activities	No exciting activities in the hour before bed
Hunger	Not going to bed hungry, or consuming a large meal before bed
Relaxed	Being relaxed and calm before bed
Location	Sleep in the same location every night
Bedroom	Sleep in a dark and quiet bedroom
Electronics	No electronics, including phones, in the bedroom
Room for sleep	Using the bed and bedroom only for sleep
Eating healthy	Eating a healthy, well-balanced diet
Physical activity	Engaging in regular physical activity

Note: Descriptors are from Allen et al., 2016 [16].

## Data Availability

Data is contained within the article or Appendix A. The data presented in this study are available in the current article and the Discussion Guide is available in the Appendix A.

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
