# Peer review of "Focus Groups to Inform User-Centered Development of an eHealth Sleep Intervention for Adolescents: Perspectives of Youth with Insomnia Symptoms, with and without Pain"

_children, 2023, doi:10.3390/children10101692_

Round 1

Reviewer 1 Report

This paper describes the first step in creating an e-health intervention to help teenagers with and without pain get better sleep. This is crucial work that addresses not only the problems with sleep that teenagers have, but also the problems with pain. I have a few minor suggestions to improve the paper.

-The discussion section should address the lack of racial diversity in the sample, the worrisome fact that nearly half of the non-pain group reported having pain.

-The discussion should address the possibility that failure "to attend focus groups despite confirming availability" led to a sampling bias since only teenagers with enough means to attend (either due to personal responsibility or outside forces) were able to give their opinions. These individuals may differ from the teenagers who did not attend. Additionally, the tendency of adolescents to not show to something online could indicate a possible pitfall for the efficacy of the eventual eHealth intervention. 

Author Response

Thank you for your feedback, which has been addressed and we believe strengthens the manuscript. 

This paper describes the first step in creating an e-health intervention to help teenagers with and without pain get better sleep. This is crucial work that addresses not only the problems with sleep that teenagers have, but also the problems with pain. I have a few minor suggestions to improve the paper.

-The discussion section should address the lack of racial diversity in the sample, the worrisome fact that nearly half of the non-pain group reported having pain.

-The discussion should address the possibility that failure "to attend focus groups despite confirming availability" led to a sampling bias since only teenagers with enough means to attend (either due to personal responsibility or outside forces) were able to give their opinions. These individuals may differ from the teenagers who did not attend. Additionally, the tendency of adolescents to not show to something online could indicate a possible pitfall for the efficacy of the eventual eHealth intervention. 

  • We have revised the Limitations section of the Discussion to be more comprehensive and to include the excellent points made by this reviewer as well as points made by the other reviewer. The additional section states – “We also were limited in describing the sample as we did not ask for information on the participant’s location within Canada and as such do not know to which geographical regions we can generalize our findings. We also did not recruit an ethnically diverse sample, and so caution needs to be taken when generalizing the study’s findings. Further caution is recommended given that had a sizable number of participants not attend the groups despite confirming their interest and availability. This could lead to a sampling bias as only those participants who had the resources (such as personal responsibility or other factors) to attend the focus groups had a voice in this study, and their opinions may differ from those who did not attend the focus groups. Moreover, this lack of engagement with online mediums may suggest that there may be uptake and adherence issues with online interventions for adolescents. We also did not ask all possible questions about eHealth interventions (e.g., we did not ask about alternatives to this such as analog versions), as we wanted to ensure that the focus groups were not too long. Lastly, we only used prompts related to pain and sleep in the pain group. It was not expected that the non-pain group would report recurrent pain on the demographic form after having not reported this on the eligibility screening questionnaire. This means that we lost an opportunity to learn from those adolescents who had recurrent pain (albeit less severe) who were in the non-pain group.”

Reviewer 2 Report

Dear authors,

This works presents a qualitative study on focus groups with adolescents and stakeholders with insomnia symptoms about healthy sleep behaviours and the input of the interviewees for the future development of an e-health intervention.

Although this work is well written, there are a few concerns to take in to account as well as a few remarks to improve the manuscript for possible publication.

·         Title: considering these focus groups were held with adolescents with insomnia symptoms it seems only right to include this in the title. Please complete the title with … insomniac adolescents with or without pain

·         Abstract line 15: associated stakeholder, please define the here as follows: i.e. parents, teachers,…

·         Line 16: 10 with pain: is that recurrent pain?

·         Line 16: online focus groups via (complete the medium; Zoom, Teams, …?)

·         Line 16: 24 adolescents with insomnia symptoms as determined by ISI (or something similar)

·         Line 20: many barriers : please define how many is many and add the barriers (i.e…..)

·         Line 24: what is ment by: ‘with accessibility as a common facilitator’, please rephrase this.

·         Line 32: add according ages after adolescents

·         Line 57: lifestyle factors, please complement with these factors (i.e. ….)

·         Line 94: in sleep outcomes, please complement with what reported outcomes  (i.e. …)

·         Line 115: mild insomnia problems, please already disclose here how insomnia was assessed; by means of ISI

·         Line 126: with and without recurrent pain? As you are including both groups, it is necessary to  include both here? Or were health care providers not included in the focus group n° 3?

·         Line 130: When were participants recruited, please add the exact period

·         Line 133: demographic information: please add all questioned variables here (i.e. or such as….)

·         Line 136: focus group n° 3: is that without health care professionals and only parents/teachers (see earlier comment), please mention here

·         Line 137: please define the used software here

·         Line 137-138: please define the exact moments/timing of the focus groups being held, in the morning/afternoon/evening/night?

·         Line 144: the role that pain may have in each topic è as you are also including a pain free group, how is pain playing a role here?

·         Line 145: audio-recorded: how were the interviews recorded? By means of recorder or smartphone

·         General concern: was an ethical committee involved in this qualitative research? If so, please clearly disclose the details in the methods section. If not, please add the rationale for conducting research without ethical approval. (the consent form mentioned on line 132 were the approved by an EC?)

·         Lines 155-157: this paragraph immerses without introduction. Please move these lines up to where it is more appropriate e.g. or add a subtitle ‘sleep’, as well as another subtitle ‘pain’

·         Line 159: ethnicity was questioned. Was location (state were the interviewee lived) questioned? If not, how are you able to generalize the findings? Or do these only apply to a single state/few states?

·         Lines 164-169: what is the rationale to include the SHI, when already used the ISI and the semi-structured interview guide. Please add the rationale in the methods.

·         Line 171: Quantitative data, such as…

·         Line 172: Descriptive statistics on…please add variables here

·         Were interviewees excluded when they knew other participants in the same focus group? Was this controlled for? And if not, please disclose in the paper whether this was or wasn’t taking into account

·         Lines 192-194: were the pain outcomes significantly different between groups? And if so, why was this not reported? And if so, please move the 6 participants with pain from the pain free to the to the pain group.

·         Lines 195-197: Why were these participants not assigned to the pain group? Please add the rational in the text.

·         Lines 220: add table 3

·         Line 220: how was this questioned, please add the semi structure interview guide in the appendix of this manuscript

·         Table 3: what references were used for these descriptions? Please add them in the legend of the table or elsewhere

·         Lines 381-390: please add a few lines on how to make sure that this eHealth interventions also makes use of electronical devices (app, smartphone, computer,…) and therefore better not be used during the evening or hours before bedtime. How will this will be taken into account when developing the e-intervention?

·         Were the interviewees ever questioned on whether they prefer an online intervention rather than an analogue version? And if not, why wasn’t this questioned?

·         Line 426: motivational components, such as….pleas complement. And references to substantiate.

·         Line 439 Canada Wide: please define how recruitments was done Canada-wide and how was this controlled for? Did you receive information on the residence of the interviewees?

·         Lines 450-451: an eHealth sleep intervention for insomniac adolescents with or without pain.

·         Line 453: health concerns: please add which concerns were meant here (i.e. …)

Author Response

Thank you for your comprehensive review of the manuscript and thoughtful comments. We have addressed all your feedback and believe that the paper is stronger having done this. We hope you find it improved and worthy of publication. 

This works presents a qualitative study on focus groups with adolescents and stakeholders with insomnia symptoms about healthy sleep behaviours and the input of the interviewees for the future development of an e-health intervention. Although this work is well written, there are a few concerns to take in to account as well as a few remarks to improve the manuscript for possible publication.

  • Title: considering these focus groups were held with adolescents with insomnia symptoms it seems only right to include this in the title. Please complete the title with … insomniac adolescents with or without pain
    • We have changed the title so that this information is now included: Focus Groups to Inform User-Centered Development of an eHealth Sleep Intervention for Adolescents: Perspectives of youth with insomnia symptoms, with and without pain
  • Abstract line 15: associated stakeholder, please define the here as follows: i.e. parents, teachers,…
    • This sentence how now been changed to “Adolescents 14-18 years old experiencing symptoms of insomnia, with and without recurrent pain, and as-sociated stakeholders (i.e., parents, school personnel, and health care providers) were recruited.”
  • Line 16: 10 with pain: is that recurrent pain?
    • Yes, we have added “recurrent” to be clear – “… (14 pain-free, 10 with recurrent pain; 10 male, 14 female).”
  • Line 16: online focus groups via (complete the medium; Zoom, Teams, …?)
    • We have not added this information here as felt that it would be more appropriate in the Methods section. Thank you for pointing out that this information was missing. It is now added to Lines 147-148
  • Line 16: 24 adolescents with insomnia symptoms as determined by ISI (or something similar)
    • We have added this information into the Abstract: “…Adolescents 14-18 years old experiencing symptoms of insomnia based on the Insomnia Sleep Index,…”
  • Line 20: many barriers : please define how many is many and add the barriers (i.e…..)
    • We added the number of barriers and briefly listed these – “Three primary barriers to sleep practices were identified, including the most common being having a variable schedule due to lifestyle factors, technology at night, academics interfering with sleep, and only in the pain group, the barrier related to pain was identified.”
  • Line 24: what is ment by: ‘with accessibility as a common facilitator’, please rephrase this.
    • We have elaborated briefly on this – “…with accessibility of the sleep information and strategies as a common facilitator.”
  • Line 32: add according ages after adolescents
    • We added the age range for the literature review that this was based on – “One in four adolescents aged 11-18 reports sleeping less than the recommended 8-10 hours a night [1]”
  • Line 57: lifestyle factors, please complement with these factors (i.e. ….)
    • We added examples of lifestyle factors – “Healthy sleep practices include following recommendations about sleep behaviours and lifestyle factors (e.g., eating healthy, exercising) [16].”
  • Line 94: in sleep outcomes, please complement with what reported outcomes  (i.e. …)
    • While there are too many outcomes to report all of them, we have added some examples to this sentence – “all reporting positive changes in sleep outcomes across a range of sleep variables (e.g., sleep efficiency) and measures [22, 34, 35].”
  • Line 115: mild insomnia problems, please already disclose here how insomnia was assessed; by means of ISI
    • We have added this information – “Adolescent participants were eligible if they were between the ages of 14 and 18 years, experienced at least mild insomnia problems on the Insomnia Severity Index (see below), …”
  • Line 126: with and without recurrent pain? As you are including both groups, it is necessary to  include both here? Or were health care providers not included in the focus group n° 3?
    • We clarified that this was for both groups of adolescents – “Adolescent (both pain and non-pain group participants) and stakeholder participants were excluded if they reported experiencing an intellectual disability, and/or visual or hearing impairment that would interfere with participation.” We also added a sentence to make it clear that adolescents and stakeholders were not in the same focus groups – “As such, adolescents and stakeholders were not in the same focus groups.” (Line 140)
  • Line 130: When were participants recruited, please add the exact period
    • The dates have been added in – “Participants were recruited between March 2017 and September 2019 from across Canada through online advertisements, social media, mailing lists, posters, and word of mouth.”
  • Line 133: demographic information: please add all questioned variables here (i.e. or such as….)
    • Given that this information is included in the measures section, we indicated to “see below” rather than repeat this information.
  • Line 136: focus group n° 3: is that without health care professionals and only parents/teachers (see earlier comment), please mention here
    • As noted above, we added the following sentence – “As such, adolescents and stakeholders were not in the same focus groups.”
  • Line 137: please define the used software here
    • We added the name of the software – “Focus groups were conducted using a secure online web conferencing software (Black-board Collaborate).”
  • Line 137-138: please define the exact moments/timing of the focus groups being held, in the morning/afternoon/evening/night?
    • We clarified the timing of the focus groups – “Each 1.5-hour focus group (all of which were in the afternoon or early evening)…”
  • Line 144: the role that pain may have in each topic è as you are also including a pain free group, how is pain playing a role here?
    • We clarified that the pain-related prompts were only included in the pain groups – “. These pain-related prompts were not included for the pain-free groups.”
  • Line 145: audio-recorded: how were the interviews recorded? By means of recorder or smartphone
    • We noted that the software recorded the interviews – “Focus groups were audio-recorded (by the conferencing software)”
  • General concern: was an ethical committee involved in this qualitative research? If so, please clearly disclose the details in the methods section. If not, please add the rationale for conducting research without ethical approval. (the consent form mentioned on line 132 were the approved by an EC?)
    • At the end of the article, consistent with the journal’s formatting requirements, there is a section titled Institutional Review Board Statement. In this section we state: The study was conducted in accordance with the Declaration of Helsinki, and approved by the Institutional Review Board (or Ethics Committee) of the IWK Health Centre (1021971; February 8, 2017). We have copied and pasted this sentence at the beginning of the Materials and Methods section.
  • Lines 155-157: this paragraph immerses without introduction. Please move these lines up to where it is more appropriate e.g. or add a subtitle ‘sleep’, as well as another subtitle ‘pain’
    • We added an introduction to this sentence, which now reads as – “Screening for sleep problems was conducted using the 7-item Insomnia Severity Index with scores ranging from 0-28, and a score of 8 or above endorsing at least mild insomnia symptoms [36]. The ISI shows high internal consistency (α = 0.90) [37].”
  • Line 159: ethnicity was questioned. Was location (state were the interviewee lived) questioned? If not, how are you able to generalize the findings? Or do these only apply to a single state/few states?
    • Participants had to live in Canada to be eligible. The sentence has been updated – “The demographic questionnaire asked participants to report age, sex, ethnicity, and geographical location being in Canada.”
  • Lines 164-169: what is the rationale to include the SHI, when already used the ISI and the semi-structured interview guide. Please add the rationale in the methods.
    • We added the rationale by adding this sentence – “This was used to describe the sample in terms of sleep habits, which is a separate construct from insomnia symptoms (as measured by the ISI).”
  • Line 171: Quantitative data, such as…
    • This was revised to include examples of quantitative data – “Quantitative data (e.g., demographic, sleep and pain questions) were analyzed using the IBM Corp. SPSS Version 24 software.”
  • Line 172: Descriptive statistics on…please add variables here
    • The sentence was re-arranged so that it was clear what the descriptive statistics were – “Quantitative data (e.g., demographic, sleep and pain questions) were analyzed using the IBM Corp. SPSS Version 24 software using descriptive statistics (i.e., frequency counts, percentages, means, standard deviations, and ranges).
  • Were interviewees excluded when they knew other participants in the same focus group? Was this controlled for? And if not, please disclose in the paper whether this was or wasn’t taking into account
    • We added a sentence to answer this question (Line 141-142) – “We did not ask nor control for whether the participants knew each other (none reported this as a concern).”
  • Lines 192-194: were the pain outcomes significantly different between groups? And if so, why was this not reported? And if so, please move the 6 participants with pain from the pain free to the to the pain group.
    • Only the pain group were asked prompts related to their pain experience and sleep. We cannot change membership of groups. We did however note this as a limitation in the Discussion. – “Lastly, we only used prompts related to pain and sleep in the pain group. It was not expected that the non-pain group would report recurrent pain on the demographic form after having not reported this on the eligibility screening questionnaire. This means that we lost an opportunity to learn from those adolescents who had recurrent pain (albeit less severe) who were in the non-pain group.”
  • Lines 195-197: Why were these participants not assigned to the pain group? Please add the rational in the text.
    • Group assignment was done at the screening phase and this was maintained throughout the study as the focus groups were different for the pain group than the non-pain group (i.e., prompts about pain and sleep were only asked in the pain group). As noted above, we have added this as a limitation to the Discussion section.
  • Lines 220: add table 3
    • Table 3 is referenced on line 239 – “Focus group moderators introduced several healthy sleep practices, see Table 3,”
  • Line 220: how was this questioned, please add the semi structure interview guide in the appendix of this manuscript
    • The semi-structured focus group discussion guide was added to the paper and can be found in Appendix A
  • Table 3: what references were used for these descriptions? Please add them in the legend of the table or elsewhere
    • A note was added to the end of the table indicating that the descriptors are from Allen et al., 2016 (Reference #16)
  • Lines 381-390: please add a few lines on how to make sure that this eHealth interventions also makes use of electronical devices (app, smartphone, computer,…) and therefore better not be used during the evening or hours before bedtime. How will this will be taken into account when developing the e-intervention?
    • We added a sentence to address this important point – “This will need to be taken into consideration when developing the eHealth intervention, as the intervention program would be available on digital devices (e.g., smartphone app, computer) and if used close to bedtime would go against recommendations for healthy sleep habits”
  • Were the interviewees ever questioned on whether they prefer an online intervention rather than an analogue version? And if not, why wasn’t this questioned?
    • We did not ask this question. The limited range of questions was added to the limitations section – “We also did not ask all possible questions about eHealth interventions (e.g., we did not ask about alternatives to this such as analog versions), as we wanted to ensure that the groups were not too long.”
  • Line 426: motivational components, such as….pleas complement. And references to substantiate.
    • We added some examples of motivational components – “Including motivational components (e.g., motivational interviewing techniques, feed-back on success with the program) within an eHealth sleep intervention is important to implement despite this suggestion being identified by only stakeholders.”
    • References were not added as this is based on feedback from the stakeholders, not from the literature
  • Line 439 Canada Wide: please define how recruitments was done Canada-wide and how was this controlled for? Did you receive information on the residence of the interviewees?
    • We reworded this sentence so that the meaning was clearer – “This online platform, however, allowed for recruitment of individuals Canada-wide, which would not have been possible if the focus groups were held in-person.”
    • We also noted the lack of geographical information as a limitation of the study – “ . We also were limited in describing the sample as we did not ask for much descriptive information including location within Canada and as such do not know to which areas we can generalize our findings across Canada.”
  • Lines 450-451: an eHealth sleep intervention for insomniac adolescents with or without pain.
    • The sentence was reworded to include this information – “This study was the first step in a user-centered approach to the development of an eHealth sleep intervention, through gathering opinions from adolescents with insomnia symptoms, some of whom had recurrent pain and others that did not, as well as stakeholders (parents, educators, and healthcare professionals).”
  • Line 453: health concerns: please add which concerns were meant here (i.e. …)
    • We added examples and rephrased to behavioural health concerns – “This study was the first step in a user-centered approach to the development of an eHealth sleep intervention, through gathering opinions from adolescents with insomnia symptoms, some of whom had recurrent pain and others that did not, as well as with and without pain and corresponding stakeholders (parents, educators, and healthcare professionals).”

Round 2

Reviewer 2 Report

Dear Authors,

You present a revised version of the manuscript. The comments were clearly answered and addressed.

Author Response

Comment: You present a revised version of the manuscript. The comments were clearly answered and addressed.

Response: Thank you, glad you felt your comments were adequately addressed.